



**Assessing sub-grid variability within satellite pixels using airborne mapping**
**spectrometer measurements**
**Wenfu Tang[1,2], David P. Edwards[2], Louisa K. Emmons[2], Helen M. Worden[2], Laura M.**
**Judd[3], Lok N. Lamsal[4,5], Jassim A. Al-Saadi[3], Scott J. Janz[4], James H. Crawford[3], Merritt**
**N. Deeter[2], Gabriele Pfister[2], Rebecca R. Buchholz[2], Benjamin Gaubert[2], Caroline R.**
**Nowlan[6]**
[1]Advanced Study Program, National Center for Atmospheric Research, Boulder, CO, USA
[2]Atmospheric Chemistry Observations and Modeling, National Center for Atmospheric Research,
Boulder, CO, USA
[3]NASA Langley Research Center, Hampton, VA 23681, USA
[4]NASA Goddard Space Flight Center, Greenbelt, MD 20771, USA
[5]Universities Space Research Association, Columbia, MD 21046, USA
[6]Harvard-Smithsonian Center for Astrophysics, Cambridge, MA 02138, USA
**Abstract**

17        Sub-grid variability (SGV) of atmospheric trace gases within satellite pixels is a key issue
in satellite design, and interpretation and validation of retrieval products. However, characterizing
this variability is challenging due to the lack of independent high-resolution measurements. Here
we use tropospheric $NO_2$ vertical column (VC) measurements from the Geostationary Trace gas
and Aerosol Sensor Optimization (GeoTASO) airborne instrument with a spatial resolution of
about 250 m × 250 m to quantify the normalized SGV (i.e., the standard deviation of the sub-grid
GeoTASO values within the sampled satellite pixel divided by their mean of the sub-grid
GeoTASO values within the sampled satellite pixel) for different satellite pixel sizes. We use the
GeoTASO measurements over the Seoul Metropolitan Area (SMA) and Busan region of South
Korea during the 2016 KORUS-AQ field campaign, and over the Los Angeles Basin, USA during
the 2017 SARP field campaign. We find that the normalized SGV of $NO_2$ VC increases with
increasing satellite pixel sizes (from ~10% for 0.5 km × 0.5 km pixel size to ~35% for 25 km × 25
km pixel size), and this relationship holds for the three study regions, which are also within the
domains of upcoming geostationary satellite air quality missions. We also quantify the temporal
variability of the retrieved $NO_2$ VC within the same satellite pixels (represented by the difference
of retrieved values at two different times of a day). For a given satellite pixel size, the temporal
variability within the same satellite pixels increases with the sampling time difference over SMA.
For a given small (e.g., <=4 hours) sampling time difference within the same satellite pixels, the
temporal variability of the retrieved $NO_2$ VC increases with the increasing spatial resolution over
the SMA, Busan region, and the Los Angeles basin.
37        The results of this study have implications for future satellite design and retrieval
interpretation, and validation when comparing pixel data with local observations. In addition, the



analyses presented in this study are equally applicable in model evaluation when comparing model
grid values to local observations. Results from the Weather Research and Forecasting model
coupled with Chemistry (WRF-Chem) model indicate that the normalized satellite SGV of
tropospheric $NO_2$ VC calculated in this study could serve as an upper bound to the satellite SGV
of other species (e.g., CO and $SO_2$) that share common source(s) with $NO_2$ but have relatively
longer lifetime.

## 1. Introduction

Characterizing sub-grid variability (SGV) of atmospheric chemical constituent fields is
important in both satellite retrievals and atmospheric chemical-transport modeling. The inability
to resolve sub-grid details is one of the fundamental limitations of grid-based models (Qian et al.,
2010) and has been studied extensively (e.g., Boersma et al., 2016; Ching et al., 2006; Denby et
al., 2011; Pillai et al., 2010; Qian et al., 2010). Pillai et al. (2010) found that the SGV of column-
averaged carbon dioxide ($CO_2$) can reach up to 1.2 ppm in global models that have a horizontal
resolution of 100 km. This is an order of magnitude larger than sampling errors that include both
limitations in instrument precision and uncertainty of unresolved atmospheric $CO_2$ variability
within the mixed layer (Gerbig et al., 2003). Denby et al. (2011) suggested that the average
European urban background exposure for nitrogen dioxide ($NO_2$) using a model of 50-km
resolution is underestimated by ~44% due to SGV.
In contrast, much less attention has been paid to the sub-grid variability within satellite
pixels (e.g., Broccardo et al., 2018; Judd et al., 2019; Tack et al., 2020). Indeed, some previous
studies (e.g., Kim et al., 2016; Song et al., 2018; Zhang et al., 2019; Choi et al., 2020) used satellite
retrievals to study SGV in models, and calculated representativeness errors of model results with
respect to satellite measurements (e.g., Pillai et al., 2010). Even though satellite retrievals of
atmospheric composition often have smaller uncertainties than model results, it has not been until
recently that the typical spatial resolution of atmospheric composition satellite products has
reached scales comparable to regional atmospheric chemistry models (< ~10 km).
Until recently, accurate in-situ measurements with sufficient spatiotemporal coverage have
not been available. As a result, it has been challenging to quantify satellite SGV, even though this
is a key issue in designing, understanding and correctly interpreting satellite observations. This is
especially important in the satellite instrument develop process, during which the required
measurement precision and retrieval resolution need to be defined in order to meet the science
goals. In addition, when validating and evaluating relatively coarse-scale satellite retrievals by
comparing with in situ observations, SGV introduces large uncertainties. This work is partly
motivated by validation requirements and considerations for the upcoming geostationary orbit
(GEO) satellite constellation for atmospheric composition that includes the Tropospheric
Emissions: Monitoring Pollution (TEMPO) mission over North America (Chance et al., 2013;
Zoogman et al., 2017), the Geostationary Environment Monitoring Spectrometer (GEMS) over
Asia (Kim et al., 2020), and the Sentinel-4 mission over Europe (Courrèges-Lacoste et al., 2017).
The measurements of the Geostationary Trace gas and Aerosol Sensor Optimization
(GeoTASO) airborne instrument provide a unique dataset for quantifying satellite SGV.
GeoTASO is an airborne remote sensing instrument capable of high spatial resolution retrieval of





UV-VIS absorbing species like $NO_2$, formaldehyde (HCHO; Nowlan et al., 2018) and sulfur
dioxide ($SO_2$; Chong et al., 2020), and with measurement characteristics similar to the GEMS and
TEMPO GEO satellite instruments. The GeoTASO data used here were taken in gapless, grid-like
patterns – or "rasters" – over the regions of interest, providing essentially continuous spatial
coverage that was repeated up to four times a day in some cases. As such, the GeoTASO data
provide a preview of the type of sampling that is expected from the GEO satellite sensors, making
the data particularly suitable for our study. We focus on the GeoTASO measurements made during
the Korea United States Air Quality (KORUS-AQ) field experiment in 2016. The measurements
from KORUS-AQ have been widely used by researchers for various air quality topics, including
quantification of emissions and model and satellite evaluation (e.g., Deeter et al., 2019; Huang et
al., 2018; Kim et al., 2018; Miyazaki et al., 2019; Spinei et al., 2018; Tang et al., 2018, 2019; Souri
et al., 2020, Gaubert et al., 2020). We further compare our findings from KORUS-AQ with flights
conducted during the NASA Student Airborne Research Program (SARP) in 2017 over the Los
Angeles (LA) Basin to test the general applicability of our findings. The KORUS-AQ mission took
place within the GEMS domain, while the SARP in 2017 is within the domain of TEMPO. Given
the similarity between the TEMPO and GEMS instruments in terms of spectral ranges, spectral
and spatial resolution, and retrieval algorithms (Al-Saadi et al., 2014), such comparison is
reasonable and useful in facilitating the generalization of the results from the study.
We use the tropospheric $NO_2$ vertical column (VC) retrieved by GeoTASO as a tool to
assess satellite SGV. $NO_2$ is an important air pollutant that is primarily generated from
anthropogenic sources such as emissions from the energy, transportation, and industry sectors
(Hoesly et al., 2018). $NO_2$ is a reactive gas with a typical lifetime of a few hours in the planetary
boundary layer (PBL), although it can also be transported over long distance in the form of
peroxyacetyl nitrate (PAN) and nitric acid. $NO_2$ is a precursor of tropospheric ozone and secondary
aerosols, and has a negative impact on human health and the environment (Finlayson-Pitts et al.,
1997). The results from this paper's analysis of $NO_2$ also have implications for other air pollutants
that share common source(s) with $NO_2$, but that have somewhat longer lifetimes, for example,
carbon monoxide (CO) and $SO_2$.
In this study, we apply a satellite pixel random sampling technique and the spatial structure
function analysis to GeoTASO data (described in Section 2) to quantify the SGV of satellite pixel
$NO_2$ VC at a variety of spatial resolutions. We analyze the relationship between satellite pixel size
and satellite SGV, and the relationship between satellite pixel size and the temporal variability of
$NO_2$ observations (Section 3). We then discuss the implications for satellite design, satellite
retrieval interpretation, satellite validation and evaluation, and satellite–in situ data comparisons
(Section 4). Implications for general local observations and grid data comparisons are also
discussed. Section 5 presents our conclusions.
**2. Data and methods**
In this section, we describe the GeoTASO instrument, campaign flights and the different
analysis techniques used to characterize the satellite pixel SGV. We outline two approaches:
satellite pixel random sampling to investigate separately both spatial variability and temporal
variability, and the construction of spatial structure functions for an alternative measure of spatial
variability.



## 2.1 GeoTASO instrument


In this study, we focus on GeoTASO retrievals of tropospheric $NO_2$ Vertical Column (VC).
GeoTASO is a hyperspectral instrument (Leitch et al., 2014) that measures nadir backscattered
light in the ultraviolet (UV; 290–400 nm) and visible (VIS; 415–695 nm). As one of NASA's
airborne UV–VIS mapping instruments, it was designed to support the upcoming GEO satellite
missions by acquiring high temporal and spatial resolution measurements with dense sampling for
optimizing and experimenting with new retrieval algorithms (Leitch et al., 2014; Nowlan et al.,
2016; Lamsal et al., 2017; Judd et al., 2019). GeoTASO has a cross-track field of view of 45° (+/-
22.5° from nadir), and the retrieval pixel size at nadir is approximately 250 m×250 m from typical
flight altitudes of 24,000–28,000 feet (7.3–8.5 km). The dense sampling of the GeoTASO datasets
is a unique feature and provides the opportunity to study the expected spatial and temporal
variability within the satellite $NO_2$ retrieval pixels at high resolution. The GeoTASO data used in
this study are mostly cloud-free. Validation of GeoTASO $NO_2$ retrievals during KORUS-AQ with
Pandora shows ~10% difference on average. The uncertainty estimate is lower than that reported
by Nowlan et al. [2016].

## 2.2 The 2016 KORUS-AQ field campaign


The KORUS-AQ field measurement campaign (Al-Saadi et al., 2014), took place in May–
June 2016, to help understand the factors controlling air quality over South Korea. One of the goals
of KORUS-AQ was the testing and improvement of remote sensing algorithms in advance of the
launches of GEMS, TEMPO, and Sentinel-4 satellite missions. It is hoped that the high-quality
initial data products from the GEO missions will facilitate their rapid uptake in air quality
applications after launch (Al-Saadi et al., 2014; Kim et al., 2020). During KORUS-AQ, GeoTASO
flew onboard the NASA LaRC B200 aircraft. We focus on the data taken over the Seoul
Metropolitan Area (SMA) that is highly urbanized and polluted, and the greater Busan region, that
is somewhat less urbanized and less polluted (Figure 1). Figure 2 shows the 12 GeoTASO data
rasters (i.e., gapless maps) acquired over SMA. Figure S1 shows the 2 GeoTASO rasters acquired
over the Busan region.

## 2.3 The 2017 SARP field campaign


During the NASA Student Airborne Research Program (SARP) flights in June 2017,
(https://airbornescience.nasa.gov/content/Student_Airborne_Research_Program), GeoTASO was
flown onboard the NASA LaRC UC-12B aircraft over the LA Basin (Figure S2, which also shows
the landcover). A detailed description and analysis of these data can be found in Judd et al. (2018;
2019). In this study, we compare our analyses and findings from KORUS-AQ with those using the
GeoTASO data over the LA Basin to test the general applicability of our findings.

## 2.4 Satellite pixel random sampling for spatial variability


GeoTASO provides continuous measurements in a gapless map pattern at high spatial
resolution (Figures 2, S1, and S2). This dataset allows us to sample and study the SGV of coarser
spatial resolution hypothetical satellite pixels sampling the same domain. To mimic satellite
observations and quantify the satellite SGV, we randomly sample the GeoTASO data with
hypothetical satellite pixels spanning 27 different pixel sizes (0.5 km×0.5 km, 0.75 km×0.75 km,





1 km×1 km, 2 km×2 km, up to 25 km×25 km). Because of the move to smaller pixel sizes in the
future satellite missions, and the limitation in the maximum hypothetical satellite pixel size
sampled using the random sampling method, the analysis of SGV only goes up to 25 km × 25 km.
This sampling process is conducted for each hour of each selected flight over the regions of interest
during the KORUS-AQ and SARP campaigns. For every sampled satellite pixel, the mean
($MEAN_{pixel}$) and standard deviation ($SD_{pixel}$) of the GeoTASO tropospheric $NO_2$ VC data within
the pixel are calculated to represent the satellite SGV. Normalized satellite SGV is calculated by
the standard deviation of the GeoTASO data within the sampled satellite pixel divided by the mean
of the GeoTASO data within the sampled satellite pixel ($SD_{pixel}/MEAN_{pixel}$).
We use a set of 10,000 hypothetical satellite pixels at each size to include all of the
GeoTASO data in the analysis and to cover as many locations as possible. Our sensitivity test
indicates that the results do not change by halving the sample size. Because the data are located
closely in space but may be sampled at slightly different times for the same flight, we separate
GeoTASO data into hourly bins for each flight before pixel sampling in order to reduce the impact
of temporal variability of the GeoTASO data within a single satellite pixel sample.
As an illustration, we describe the procedure below for the May 17th afternoon flight
(Figure 3) that was conducted from 13:00 to 17:00 local time: (1) the GeoTASO data during this
flight were divided into four hourly groups according to the measurement time, i.e., 13:00-14:00,
14:00-15:00, 15:00-16:00, and 16:00-17:00; (2) for each of the 27 hypothetical satellite pixel sizes,
we randomly generate 10,000 satellite pixel locations within each hourly group. Therefore, for
each hour, we sample 270,000 satellite pixels (27 different satellite pixel sizes and 10,000 samples
for each size), and for this example flight, we have a total of up to 1,080,000 possible satellite
pixels in each of 4 hourly groups. Note that the actual samples used in the analysis are less than
1,080,000 because we discarded a sampled satellite pixel if it is not covered by GeoTASO data for
at least 75% of its area.
We tested other choices of the coverage threshold in addition to 75% over SMA (not shown
here). The results are similar for small pixels ($< \sim 10$ $km^2$), as they are more likely to be covered
by GeoTASO data regardless of the threshold value. For larger pixels ($> \sim 15$ $km^2$), the satellite
SGV is slightly lower when using 30% or 50% as the area coverage threshold, because larger
pixels act like smaller pixels when only partially covered. The threshold of 75% was chosen as a
trade-off between sample size and representation.
**2.5 Satellite pixel random sampling for temporal variability**
We also quantify the temporal variability of the retrieved $NO_2$ VC within the same satellite
pixels for different satellite pixel sizes. To calculate temporal variability within a hypothetical
satellite pixel, we need GeoTASO data to cover the hypothetical satellite pixel at different times
during the day. During the KORUS-AQ and 2017 SARP campaigns, rasters were treated as single
units (Judd et al., 2019). Each raster produces a contiguous map of data that we consider as roughly
representative of the mid-time of the raster. Unlike the calculation of SGV, which is based on data
separated into hourly bins (section 2.4) to reduce the impact of temporal variability in the
calculated spatial variability, the satellite pixel random sampling to assess temporal variability is
based on rasters, and only conducted for days with multiple rasters. This is to ensure that the





sampled hypothetical satellite pixels have multiple values at different times of the day. and hence
maximize the sample size.
To assess temporal variability within the hypothetical satellite pixels, we randomly select
50,000 pixel locations for each of the 27 hypothetical satellite pixel sizes, and use this same set of
pixel locations to sample the GeoTASO data for each raster across all flights for a given day. This
process is repeated for all days with multiple rasters, and the 75% of area coverage threshold is
also applied. When there are two or more raster values of $\text{MEAN}_{\text{pixel}}$ for a given pixel location
separated by time Dt, the temporal mean difference (TeMD) within the satellite pixel is calculated
as:
$$\text{TeMD(Dt)} = average(\,|\text{MEAN}_{\text{pixel}}(t) - \text{MEAN}_{\text{pixel}}(t+Dt)|\,) \qquad (1)$$
This procedure is repeated for each satellite pixel size.
**2.6 Spatial structure function**
Structure functions have been applied to in situ measurements and model-generated
tropospheric trace gases to analyze their spatial and temporal variability in previous studies (Harris
et al., 2001). The Spatial Structure Function (SSF) (Fishman et al., 2011; Follette-Cook et al., 2015)
is an alternative measure to the satellite pixel random sampling described above for quantifying
spatial variability, and in this work, we apply the SSF to GeoTASO data to assist our analysis of
satellite SGV. The main difference between the two measures is that the SSF is based on individual
GeoTASO data points, while the results from satellite pixel random sampling are based on sampled
satellite pixels. The SSF is defined here follows Follette-Cook et al. (2015):
$$f(NO_{2,VC}, D) = average(\,|NO_{2,VC}(x+D) - NO_{2,VC}(x)|\,) \qquad (2)$$
where $NO_{2,VC}$ is tropospheric $NO_2$ VC. $f(NO_{2,VC}, Distance)$ calculates the average of the
absolute value of $NO_{2,VC}$ differences across all data pairs (measured in the same hourly bin) that
are separated by a distance $D$. To calculate SSF, the first step is the same as the first step of the
satellite pixel random sampling: we group GeoTASO data hourly for each flight to reduce the
impact of temporal variability of the GeoTASO data, and we only pair each GeoTASO data point
with all the other GeoTASO data in the same hourly bin. More details on structure functions can
be found in Follette-Cook et al. (2015).
**2.7 WRF-Chem simulation**
To briefly demonstrate the application of this technique on model evaluation and other
species, we show results of a WRF-Chem simulation (Weather Research and Forecasting model
coupled to Chemistry) with a resolution of 3 km × 3 km over SMA in the Discussion section. The
simulation used NCEP GDAS/FNL 0.25 Degree Global Tropospheric Analyses and Forecast Grids
as initial and boundary conditions, and the model meteorological fields above the PBL were
nudged 6-hourly. KORUS version 3 anthropogenic emissions and FINN version 1.5 fire emissions
(Wiedinmyer et al., 2011) were used.





**3. Results**

In this section, we discuss the results for SGV over the different regions considered. Results are presented for the hypothetical satellite pixel random sampling for spatial variability and temporal variability, and for the spatial structure function analysis for spatial variability.

**3.1 Sub-grid variability (SGV) within satellite pixels**

SMA, the Busan region, and the LA Basin have different levels of pollution – the average values of the GeoTASO $NO_2$ VC data over the SMA, the Busan region, and the LA Basin are $2.3\times10^{16}$ molecules $cm^{-2}$, $1.1\times10^{16}$ molecules $cm^{-2}$, and $1.3\times10^{16}$ molecules $cm^{-2}$, respectively. Over the three regions, the mean values ($MEAN_{pixel}$) and absolute values of standard deviation ($SD_{pixel}$) of the hypothetical satellite pixels sampled over GeoTASO $NO_2$ VC data are different (Figure S3). This is consistent with previous studies suggesting absolute values of SGV can vary regionally (Judd et al., 2019; Broccardo et al., 2018). However, we find that the normalized satellite SGV (calculated as the ratio of $SD_{pixel}$ to $MEAN_{pixel}$ for a sampled pixel) is similar over each of the areas, regardless of the absolute level of pollution as represented by $MEAN_{pixel}$ (Figure 4). Over SMA (Figure 4a), the mean normalized satellite SGV of tropospheric $NO_2$ VC increases smoothly from ~10% for the pixel size of 0.5 km × 0.5 km, to ~35% for the pixel size of 25 km × 25 km. The interquartile variation of the satellite SGV also increases with satellite pixel sizes. The patterns of the sampled satellite pixels over the Busan region (Figure 4b) and LA Basin (Figure 4c) are also found to be similar to those over SMA. Furthermore, Figures S4 and S5 show that even the individual flights over the three domains generally follow the same pattern, except in the case of the June 9 PM flight that is discussed below.

We also compare normalized satellite SGV for different levels of pollution, regardless of their regions (Figure S6). The normalized satellite SGV for the less polluted pixels ($MEAN_{pixel}$ being lower than the average value of all pixels, i.e., $2\times10^{16}$ molecules $cm^{-2}$) also shows an overall similar pattern as for the more polluted pixels ($MEAN_{pixel}$ being higher than the average value of all pixels). We notice that at small pixel sizes, less polluted pixels have higher normalized satellite SGV, possibly contributed by relatively higher retrieval noise at lower pollution levels.

In addition to the comparison between different domains and pollution levels, we also compare this relationship in the morning and afternoon. The variation of normalized SGV and pixel size in the morning and afternoon are generally similar for the three regions (Figure S7), except for the large size pixels over SMA, where the normalized SGV is larger in the afternoon than in the morning. This difference is driven by the GeoTASO data from June 9 PM (Figure S4), as the normalized SGV pattern for the afternoon agrees well with the normalized SGV pattern for the morning when the June 9 PM data are excluded. Figure S1 shows that the June 9 PM $NO_2$ pollution level is higher than other days under meteorological conditions of light winds and moderate temperatures. The $MEAN_{pixel}$ values increases ~60% going from 1 km × 1 km to 25 km × 25 km pixel size, while $SD_{pixel}$ dramatically increases ~7 times from 1 km × 1 km to 25 km × 25 km. This is higher than any other day, and results in the highest SGV encountered over SMA at the large pixel sizes. We also notice that the normalized SGV does not generally change significantly in the range of 20 km × 20 km to 25 km × 25 km. However, in the case of SMA for June 9 PM, the normalized SGV (as well as $SD_{pixel}$) increases significantly and monotonously with pixel size in the range of 20 km × 20 km to 25 km × 25 km.



We show the normalized SGV for individual rasters over SMA (Figure 5) to indicate the
uncertainty range of the normalized SGV shown in Figure 4. The spread of SGV across different
individual rasters represents the uncertainties of using the averaged normalized SGV for a specific
case. Note that the variation of normalized SGV with pixel size for individual rasters generally
follows the same pattern (i.e., increases with satellite pixel size), especially when the pixel size is
small (≤10 km × 10 km). The normalized SGV increases from ~10% to ~25%, with the uncertainty
range consistently being ±5% when the pixel size is smaller than 10 km × 10 km. When the pixel
size is larger than 10 km × 10 km, the uncertainty range broadens with pixel sizes from ±5% (10
km × 10 km) to ±15% (25 km × 25 km). This means that when the satellite pixel size is large,
using the mean normalized SGV in Figure 4 to represent specific cases may lead to larger
uncertainties. Therefore, our analysis reveals a threshold for spatial resolution at about 10 km × 10
km. Below this resolution, SGV can be characterized by the mean value with relatively smaller
uncertainty (±5%) and hence high confidence, even with large diurnal or day-to-day variations.
The spatial resolutions of TEMPO, GEMS, and TROPOMI (TROPOspheric Monitoring
Instrument, Veefkind et al., 2012; Griffin et al., 2019; van Geffen et al., 2019) are within this ≤10
km × 10 km range, while the resolution of OMI (Ozone Monitoring Instrument, Levelt et al., 2006;
2018) is not. This means that applying this study (e.g., Figure 4) to OMI for a specific case study
(e.g., a specific day) requires extra caution.
We tested the sensitivity of the results over SMA to sampling GeoTASO data with
hypothetical satellite pixels grouped by complete flight, rather than grouping the data by time in
hourly bins. The resulting patterns and relationships are similar, except that the normalized satellite
SGV increases ~5% for pixels of small sizes due to the inclusion of temporal variability (Figure
S8a). We also tested the results for sampling satellite pixels by raster instead of within hourly bins.
The results are again similar to Figure 4, except that the normalized satellite SGV increases ~1%
for pixels of small sizes due to the inclusion of temporal variability (Figure S8b).
The three regions investigated in this work have different levels of urbanization and air
pollution (Figures 1 and S2). PBL conditions are also different in the morning and afternoon
(Figure S9). The similarity of the relationships between the satellite pixel size and the normalized
satellite SGV over these different regions (Figure 4) suggests that this relationship may be
generalizable to $NO_2$ VC over regions with different levels of urbanization and air pollution, and
different PBL conditions. Moreover, Figures 4 and 5 point to the possibility of developing a
generalized look-up table for the expected normalized satellite SGV for $NO_2$ VC at different
satellite pixel sizes, especially for small pixel sizes (e.g., TEMPO, GEMS, and TROPOMI). This
would be useful in satellite design, satellite retrieval evaluation and interpretation, and satellite–in
situ data comparisons. For example, the satellite pixel size of tropospheric $NO_2$ VC retrievals from
GEMS, TEMPO, TROPOMI, and OMI are highlighted in Figure 4. Following Judd et al. (2019),
we choose 3 km × 3 km, 5 km × 5 km, 7 km × 8 km, and 18 km × 18 km pixels to represent the
expected area of the satellite pixels for TEMPO (2.1 km × 4.4 km), TROPOMI (3.5 km × 7 km),
GEMS (7 km × 8 km), and OMI (18 km × 18 km), respectively. The expected normalized satellite
SGV for TEMPO, TROPOMI, GEMS, and OMI are 15–20%, ~20%, 20–25%, and ~30%,
respectively. Taking the TEMPO example, this implies that the satellite SGV could potentially
lead to uncertainties of 15–20% in a validation exercise comparing a satellite retrieval with sub-
satellite local ground measurements of $NO_2$ VC as might be obtained from a Pandora spectrometer.



As a result, we should caution that calculating a pixel mean bias when evaluating against local
measurements within the pixel sometimes may be optimistic due to the cancellation of sub-grid
positive and negative biases.
**3.2 Temporal variability (TeMD) within the same satellite pixels**
In addition to satellite spatial SGV, we also analyze the temporal variability (i.e., TeMD)
within the same hypothetical satellite pixels. Figure 6 shows TeMD of satellite retrieved
tropospheric $NO_2$ VC over SMA as a function of hypothetical satellite pixel size and the separation
time Dt between flight rasters as described in section 2.5. The results for 27 satellite pixel sizes
analyzed are shown by different colors, while results for selected satellite pixel sizes are
highlighted by thicker lines. For all the pixel sizes, TeMD increases monotonically with the time
difference Dt between two sampled raster values within the same pixel. The TeMD of tropospheric
$NO_2$ VC is around $0.75\times10^{16}$ molecules $cm^{-2}$ for a Dt of 2 hours over SMA for all the sampled
satellite pixel sizes, and increases to $\sim2\times10^{16}$ molecules $cm^{-2}$ for Dt of 8 hours. This indicates that,
along with improvements in the satellite retrieval spatial resolution with smaller pixels, improving
the satellite retrieval temporal resolution with higher frequency measurements is also an effective
way to enhance capability in resolving variabilities of $NO_2$. This is expected because of $NO_2$'s
relatively short lifetime (~ a few hours) and strong diurnal cycle due to emission activities,
chemistry and photolysis rate (Fishman et al., 2011; Follette-Cook et al., 2015). The diurnal cycle
of the PBL also plays a large role because horizontal dispersion occurs as the PBL thickens during
the day. Early in the morning, the PBL is low (~1400 m during 9:00-11:00 in SMA) and strong
sources are evident such as traffic on major highways, etc. As the day progresses, the PBL height
increases (~1800 m during 15:00-17:00; Figure S9) allowing for greater horizontal mixing to take
place. By early afternoon, emissions from all the major sources in the central region have mixed
together to form a wide area of high pollution over the urban center. Judd et al. (2018) point out
that the topography over SMA also plays a role in the ability to mix horizontally as the PBL grows.
Therefore, the TeMD can be large between morning and afternoon (i.e., for Dt larger than 6 hours).
For a small Dt (2 or 4 hours), TeMD increases when increasing the satellite retrieval spatial
resolution (i.e., smaller pixel size). This is especially true for short time periods (e.g., 2 hours and
4 hours), which is more important for the GEO satellite measurements. For example, for Dt of 2
hours, TeMD for satellite pixels of 1 km × 1 km is about $0.80\times10^{16}$ molecules $cm^{-2}$, while TeMD
for satellite pixels of 25 km × 25 km is about $0.73\times10^{16}$ molecules/$cm^2$ (~9% lower); when Dt is
4 hours, TeMD for satellite pixels of 1 km × 1 km is about $1.3\times10^{16}$ molecules $cm^{-2}$, while TeMD
for satellite pixels of 25 km × 25 km is about $1.1\times10^{16}$ molecules/$cm^2$ (~15% lower). This indicates
that when increasing the satellite retrieval spatial resolution (decreasing pixel size), the temporal
variability of the retrieved values will increase, even though the normalized satellite spatial SGV
decreases. Thus, temporal resolution should be increased in conjunction with the increase in spatial
resolution in order to enhance the accuracy of the satellite products. This is expected because
averaging over a larger region smooths out temporal variability so producing smaller hourly
differences. Our finding here is consistent with that of Fishman et al. (2011).
GeoTASO data over the Busan region is limited. Given the fewer flights, we are not able
to show how TeMD changes with Dt over the Busan region in this study. However, we are able to
show the relationship between TeMD and satellite pixel sizes for a limited range of Dt. During
KORUS-AQ, there were only two rasters sampled over Busan with a Dt of 2 hours (Figure S10).



For this Dt of 2 hours, TeMD increases slightly when increasing the satellite retrieval spatial
resolution (smaller pixel size). More data over the Busan region would help significantly for this
analysis. As for sampled hypothetical satellite pixels over the LA Basin, for a given Dt, TeMD
increases when increasing the satellite retrieval spatial resolution (smaller pixel size) for Dt equal
to 4 and 8 hours (Figure S11). We note that with only 2 flight days of flight data, the GeoTASO
data over LA is also limited. Despite the limited sample sizes, TeMD increases when increasing
the satellite retrieval spatial resolution over both the Busan region and the LA Basin, which is
consistent with the relationships over the SMA for a small Dt.
**3.3 Results from Spatial Structure Function (SSF)**
In this section, we show the analysis of SSF over SMA (Figure 7) as a complement to our
analysis in Section 3.1. As mentioned before, SSF and SGV are different measures of spatial
variability and are not directly comparable. This is because SSF is calculated based on differences
between a single GeoTASO measurement and all the other GeoTASO measurements on the map,
while SGV is derived based on variation among all the GeoTASO measurements within a
hypothetical satellite pixel unit. SSF measures the averaged spatial difference at a given distance,
while SGV directly quantifies the expected spatial variability within a satellite pixel at a given size.
As both SSF and SGV are related to spatial variability, we include SSF in this study as an extension
to SGV.
Figure 7a shows that the SSF in SMA initially increases with the distance between data
points, peaks at around 40-60 km during most flights, and then decreases with distance between
60 and 140 km. The number of paired GeoTASO data points when the distance is larger than 100
km is relatively small (Figure S12) therefore conclusions beyond this distance are not included in
this analysis. The increases in SSF for distances in the range of 1-25 km (Figure 7b) are consistent
with the relationship between pixel sizes and the normalized satellite SGV shown in Figure 4. For
example, over the 1-25 km range, Fig 4a shows the median increases from around 8% to around
28%, an increase by a factor of 3.5, and the black line in Figure 7 shows an approximately similar
factor (from $0.33 \times 10^{16}$ molecules/cm$^2$ for 1 km to $1.5 \times 10^{16}$ molecules/cm$^2$ for 25 km). This
increase of SSF between 1-25 km is also seen over the Busan region and the LA Basin (Figure
S13). We also notice that SSF shows a relatively strong dependence on the particular GeoTASO
flight, while SGV is less sensitive, especially for small pixel sizes.
The shapes of the SSF are generally consistent with previous studies for modeled or in situ
observations of $NO_2$ (Fishman et al., 2011; Follette-Cook et al., 2015). Previous studies also
suggest that different aircraft campaigns may share the common shape of SSF but different
magnitudes, which is strongly related to the fraction of polluted samples versus samples of
background air in the campaign (Crawford et al., 2009; Fishman et al., 2011). Differences in the
shape and size of particular cities also contribute to the differences in the SSF. For example, at a
certain distance SSF may compare polluted areas within the same urban region, while over a
different smaller city, the comparison at the same distance reveals the gradient between the
polluted city and cleaner surrounding background air, so resulting in different peak values. Valin
et al. (2011) found that the maximum in OH feedback in a NOx-OH steady-state relationship
corresponds to a $NO_2$ e-folding decay length of 54 km in 5m/s winds. This may partially explain
the peak between 40~60 km in SSF. As shown in Figures 2 and S7, the overall spatial variability
over SMA is higher in the afternoon. Over SMA, the SSF in the morning is generally smaller than





in the afternoon, indicating higher spatial variability of tropospheric $NO_2$ VC in the afternoon (see
also Judd et al., 2018). As described in Section 2.6, SSF discussed here (Figure 7) is calculated
based on hourly bin. We also include SSF that is calculated within rasters in the supplement (Figure
S14). The overall shapes of SSF (Figure S14) calculated on raster basis are similar to SSF
calculated on hourly basis (Figure 7).
Previous studies (Fishman et al., 2011; Follette-Cook et al., 2015) used SSF values at a
particular distance to indicate the satellite precision requirement at a corresponding resolution in
order to resolve spatial structure over the pixel scale. For GEMS, the expected spatial differences
over the scale of its pixel for the SMA and Busan regions are ~$7.5\times10^{15}$ molecules cm$^{-2}$ and
~$3.5\times10^{15}$ molecules cm$^{-2}$, respectively, taking the SSF values at 5 km to be representative. For
TEMPO, the spatial difference is ~$2.8\times10^{15}$ molecules cm$^{-2}$ over LA Basin taking the SSF value
at 3 km. Assuming the $NO_2$ measurement precision requirement to be $1\times10^{15}$ molecules cm$^{-2}$ for
both TEMPO and GEMS (Chance et al., 2013; Kim et al., 2020), the expected spatial differences
over the three regions are considerably higher than the precision requirement and should be easily
characterized by both the GEMS and TEMPO missions.
**4. Discussions and implications**
The relationship between satellite pixel sizes and the normalized satellite SGV is fairly
robust over the different regions studied here, and Figure 4 points to the possibility of developing
a generalized look-up table if more data were available in other regions. A generalized relationship
between satellite pixel sizes and the temporal variability (Figure 6) is not as evident as the
relationship between satellite pixel sizes and the normalized satellite SGV due to limited data.
However, it is still useful for satellite observations over SMA, which is in the GEMS domain and
should be helpful in satellite retrieval interpretation.
This study also has implications for satellite validation and evaluation, and satellite–in situ
data comparisons of other trace gas species. Our initial motivation to study satellite SGV arose
from our previous work on validation of MOPITT (Measurements of Pollution in the Troposphere)
CO retrievals over urban regions (Tang et al., 2020). In that study, we compared the satellite
retrievals with aircraft profiles, and realized that satellite SGV and representativeness error of
aircraft profiles in the comparisons to MOPITT retrievals introduced uncertainties in the validation
results. Previous studies have noticed the same issue for $NO_2$ (e.g., Nowlan et al., 2016, 2018;
Judd et al., 2019; Pinardi et al., 2020; Tack et al., 2020), but this issue is difficult to address and
quantify due to the limited spatial coverage of most aircraft observations. Even though only a few
trace gas species are routinely retrieved, the gapless raster datasets of GeoTASO are a possible
way to address this problem. The normalized SGV of the GeoTASO tropospheric $NO_2$ VC might
serve as an upper bound to the SGV of CO, $SO_2$ and other species that share common source(s)
with $NO_2$ but have relatively longer lifetimes, even if their spatial distributions may have different
patterns (e.g., Chong et al., 2020). For example, at the resolution of 22 km × 22 km (resolution of
MOPITT CO retrievals), the expected normalized satellite SGV of tropospheric $NO_2$ VC is ~30%.
Therefore, we might expect the normalized satellite SGV for tropospheric CO VC to be lower than
this value.
To demonstrate this idea, we use the WRF-Chem regional model at an intermediary step.
At the model resolution, if the SVG of the WRF-Chem model and GeoTASO $NO_2$ VC agree





reasonably well, then the model can be used to predict the SVG of other species that are chemically
constrained with $NO_2$ at the model resolution and at coarser resolutions. This is shown in Figure 8
which illustrates how SGV varies with satellite pixel size for $NO_2$ VC, CO VC, $SO_2$ VC, and
formaldehyde (HCHO) VC calculated from a WRF-Chem simulation. The modeled $NO_2$, CO, $SO_2$,
and HCHO concentrations are converted to VC, and are filtered to match the rasters of GeoTASO
measurements (Figure S15). As expected, SGV of modeled $NO_2$ VC is higher than SGV of
modeled CO VC, $SO_2$ VC, and HCHO VC. We also notice that SGV for modeled $NO_2$ VC, CO
VC, $SO_2$ VC, and HCHO VC increases with pixel size, which is similar to that for GeoTASO
measurements. The SGV for GeoTASO $NO_2$ shown in this figure (black lines) is calculated based
on GeoTASO data that are regridded to the WRF-Chem grid (3 km × 3 km), making it slightly
different from that in Figure 4. Note that a more comprehensive comparison requires further work
and ideally actual dense GeoTASO-type measurements of CO and other species to address
differences due to local sources on the background concentrations.
This study is also relevant to model comparison and evaluation with local observations.
Whenever local observations are compared to grid data (e.g., comparisons between satellite
retrievals and local observations, comparisons between grid-based model and local observations,
and data assimilation), SGV will introduce uncertainties that need to be quantified to better
interpret and understand the comparison results. For example, we note that at the resolution of 14
km×14 km (a typical resolution for the forward-looking Multi-Scale Infrastructure for Chemistry
and Aerosols Version 0; MUSICA-V0, https://www2.acom.ucar.edu/sections/multi-scale-
chemistry-modeling-musica; Pfister et al. [2020]), the expected normalized satellite SGV of
tropospheric $NO_2$ VC is ~25-30%. When comparing model simulations at a coarser resolution with
local observations for tropospheric $NO_2$ VC, a normalized SGV larger than ~25-30% may be
expected. If comparing for a specific vertical layer instead of vertical column, an even larger
normalized SGV may occur.
**5. Conclusions**
Satellite SGV is a key issue in interpreting satellite retrieval results. Quantifying studies
have been lacking due to limited high-resolution observations. In this study, we quantified likely
GEO satellite SGV by using GeoTASO measurements of tropospheric $NO_2$ VC over the urbanized
and polluted Seoul Metropolitan Area (SMA) and the less-polluted Busan region during KORUS-
AQ, and the Los Angeles (LA) Basin during the 2017 SARP campaigns. The main findings of this
work are the following:
(1) The normalized satellite SGV increases with hypothetical satellite pixel sizes based on satellite
pixel random sampling of hourly GeoTASO data, from ~10% (±5% for specific cases such as
an individual day/time of day) for a pixel size of 0.5 km × 0.5 km to ~35% (±10% for specific
cases such as an individual day/time of day) for the pixel size of 25 km × 25 km. This
conclusion holds for all the three study regions, despite their different levels of urbanization
and pollution, and for time of day, morning or afternoon.
(2) The normalized satellite SGV of tropospheric $NO_2$ VC could serve as an upper bound to
satellite SGV of CO, $SO_2$ and other species that share common source(s) with $NO_2$ but have
relatively longer lifetime, as supported by the high-resolution WRF-Chem simulation.
(3) The temporal variability (TeMD) within the same hypothetical satellite pixels increases with
sampling time differences (Dt) over SMA. TeMD ranges from ~$0.75×10^{16}$ molecules cm$^{-2}$ at
Dt of 2 hours to ~$2\times10^{16}$ molecules $cm^{-2}$ (about three times higher) at Dt of 8 hours. TeMD is
likely impacted by the short lifetime and diurnal cycle of $NO_2$ due to emission activities and
photolysis rate, and the meteorology and PBL evolution during the day. Improving the satellite
retrieval temporal resolution is an effective way to enhance the capability of satellite products
in resolving variabilities of $NO_2$.
(4) Temporal variability (TeMD) increases when increasing the satellite retrieval spatial resolution
(i.e., smaller pixel size) in SMA. For example, when Dt is 2 hours, TeMD for satellite pixels
with the size of 25 km × 25 km is about 20% lower compared to TeMD for satellite pixels with
the size of 1 km × 1 km. Thus, temporal resolution should be increased along with any increase
in spatial resolution in order to enhance the accuracy of satellite products.
(5) The spatial structure function (SSF) firstly increases with the distance between data points,
peaks at around 40-60 km during most flight days, and then decreases with distance. This is
generally consistent with previous studies.
(6) SSF analyses suggest that GEMS will encounter $NO_2$ VC pixel scale spatial differences of
~$7.5\times10^{15}$ and ~$3.5\times10^{15}$ molecules $cm^{-2}$ over the SMA and Busan regions, respectively.
TEMPO will encounter $NO_2$ VC spatial differences at its pixel scale of ~$2.8\times10^{15}$ molecules
$cm^{-2}$ over the LA Basin. These differences should be easily resolved at the stated measurement
precision requirement of $1\times10^{15}$ molecules $cm^{-2}$.
(7) These findings are relevant to future satellite design and satellite retrieval interpretation,
especially now with the deployment of the high-resolution GEO air quality satellite
constellation, GEMS, TEMPO, and Sentinel-4. This study also has implication for satellite
product validation and evaluation, satellite–in situ data comparisons, and more general point-
grid data comparisons. These share similar issues of sub-grid variability and the need for
quantification of representativeness error.
We note that this study has some uncertainties and limitations. (1) The variability at a
resolution finer than 250 m × 250 m (i.e., GeoTASO's resolution) may introduce uncertainties to
the analysis here, although this is beyond the scope of this study. (2) Even though a large number
of GeoTASO retrievals have been analyzed in this study, we would still benefit from more high-
resolution measurements with a broader spatiotemporal coverage, particularly over the Busan
region. More GeoTASO-type data over the Busan region will help testing the consistence in TeMD
over different regions. (3) The KORUS-AQ campaign was conducted in Spring (May and June),
and the 2017 SARP campaign was also conducted in June. More GeoTASO-type measurements
over South Korea during different season(s) would be particularly helpful to understand and
generalize the findings in this study.
This work demonstrates the value of continued flights of GeoTASO-type instruments obtaining
continuous, high spatial resolution data several times a day, particularly for the upcoming
validation exercises for the GEO air quality satellite constellation.

**Acknowledgement**
The authors thank the GeoTASO team for providing the GeoTASO measurements. The authors
thank the KORUS-AQ and SARP team for the campaign data. We thank the DIAL-HSRL team





for the mixing layer height data (available at https://www-air.larc.nasa.gov/cgi-bin/ArcView/korusaq). Tang was supported by a NCAR Advanced Study Program Postdoctoral Fellowship. Edwards was partially supported by the TEMPO Science Team under Smithsonian Astrophysical Observatory Subcontract SV3-83021. The National Center for Atmospheric Research (NCAR) is sponsored by the National Science Foundation. The authors thank Ivan Ortega and Sara-Eva Martinez-Alonso for helpful comments on the paper.

**Data availability**
The KORUS-AQ and SARP data are available at https://www-air.larc.nasa.gov/cgi-bin/ArcView/korusaq and https://www-air.larc.nasa.gov/cgi-bin/ArcView/lmos, respectively.

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

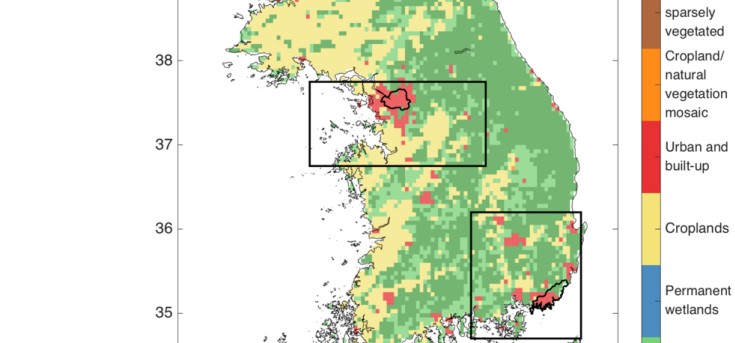

**Figure 1**. Domain of the study over South Korea and the land cover. Boxes indicate location of
the SMA (upper left) and the Busan region (lower right) domains. Land cover data are from
MODIS Terra and Aqua MCD12C1 L3 product, version V006, annual mean at 0.05° resolution;
Friedl et al., 2015.






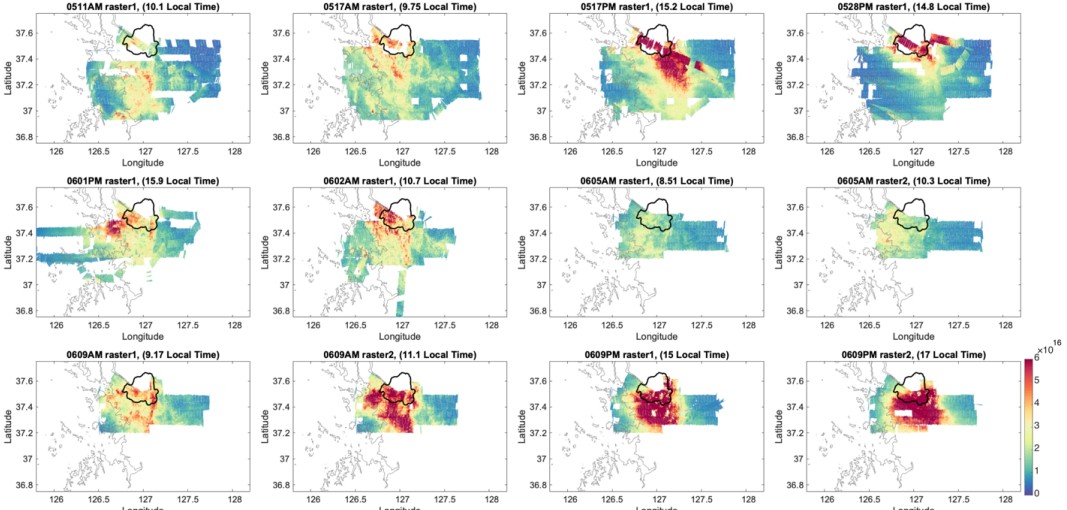


**Figure 2**. GeoTASO data of tropospheric $NO_2$ vertical column (molecules $cm^{-2}$) measured during
KORUS-AQ over the Seoul region. Each panel shows a separate raster. Panel titles show month,
day, AM/PM, raster number on that date, and mean time of raster acquisition. There were nine
flights sampling rasters over Seoul. The May 01 AM, May 17 AM, May 17 PM, May 28 PM, June
01 PM, and June 02 AM flights each sampled one raster. The June 05 AM, June 09 AM, and June
09 PM flights each sampled two rasters. As a result, there were two flights and two rasters on May
17th, one flight and two rasters on June 5th, and two flights and four rasters on June 9th.







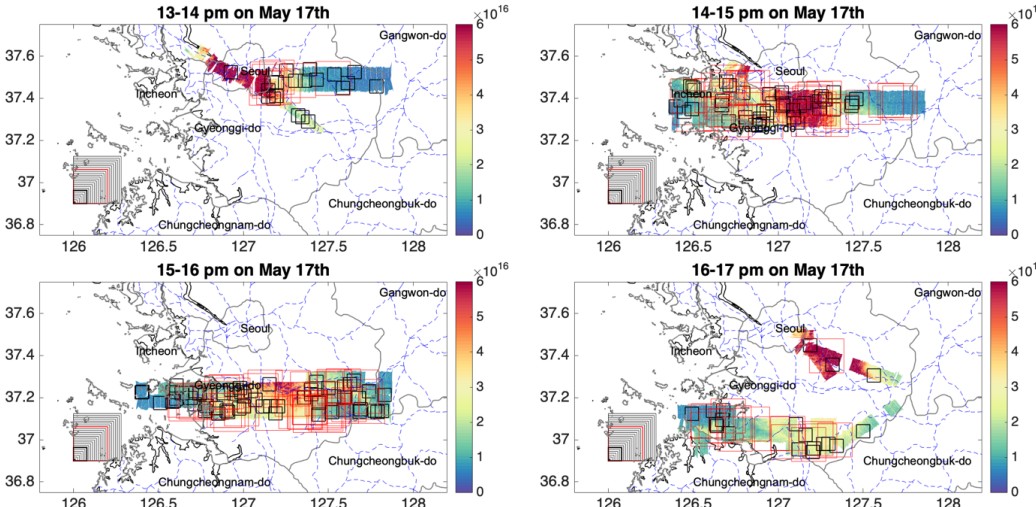

**Figure 3**. Demonstration of the hypothetical satellite pixel random sampling method. Each subplot
is an hour during May 17th PM flight. For each hour, we randomly sample 10000 hypothetical
satellite pixels at each different pixel sizes (i.e., 0.5 km×0.5 km, 0.75 km×0.75 km, 1 km×1 km, 2
km×2 km, … , 25 km×25 km) over the GeoTASO data of tropospheric $NO_2$ vertical column
(molecules $cm^{-2}$) every hour. The sampled pixel size (from 0.5 km×0.5 km to 25 km×25 km) are
shown in the lower-left corner of each sub-plot. Only 100 samples for pixel size of 7 km×7 km
(thick black box) and 100 samples for 18 km × 18 km are shown for demonstration purposes.
Samples that fail to pass the 75% coverage threshold are not shown. Coastlines,
Province/Metropolitan City boundaries are shown by gray solid lines. Main roads are shown by
blue dashed lines (data are from http://www.diva-gis.org/gdata).






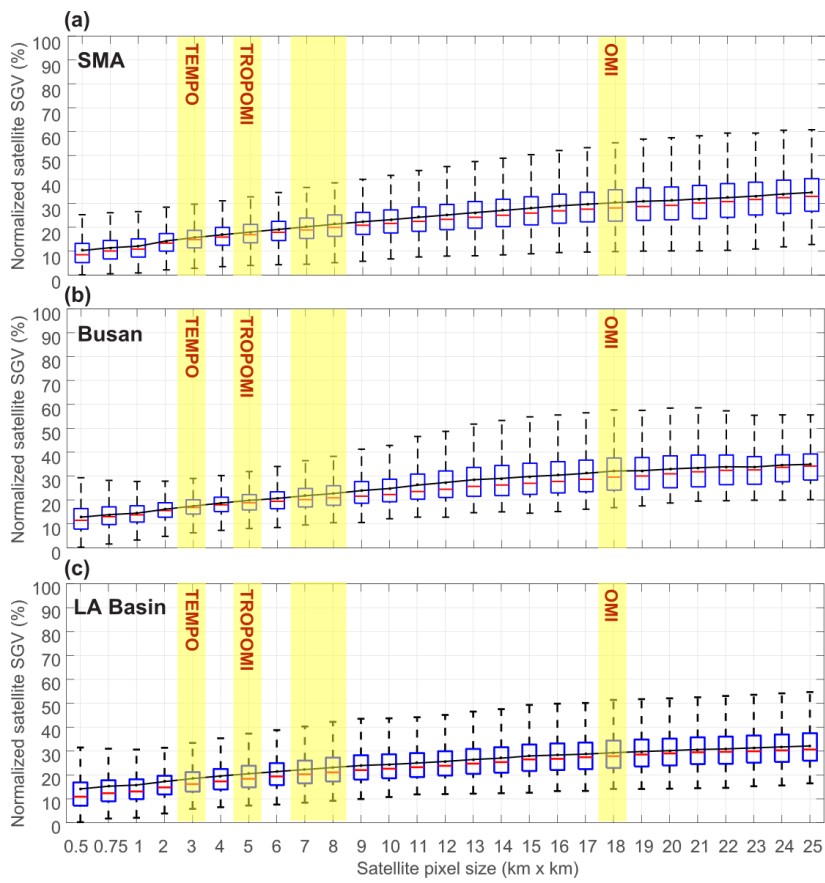

**Figure 4**. Boxplot (with medians represented by red bars, interquartile ranges between 25th and
75th percentiles represented by blue boxes, and the most extreme data points not considered
outliers represented by whiskers) for the normalized satellite sub-grid variability (SGV) over the
Seoul Metropolitan Area (a), the Busan region (b), and Los Angeles Basin (c). Normalized satellite
SGV is calculated as the standard deviation of the GeoTASO data within the sampled satellite
pixel divided by the mean of the GeoTASO data within the sampled satellite pixel. The black lines
represent the mean of the normalized satellite SGV at a given size. The resolutions of TEMPO,
TROPOMI, GEMS, and OMI are highlighted by the yellow shade in the Figure.

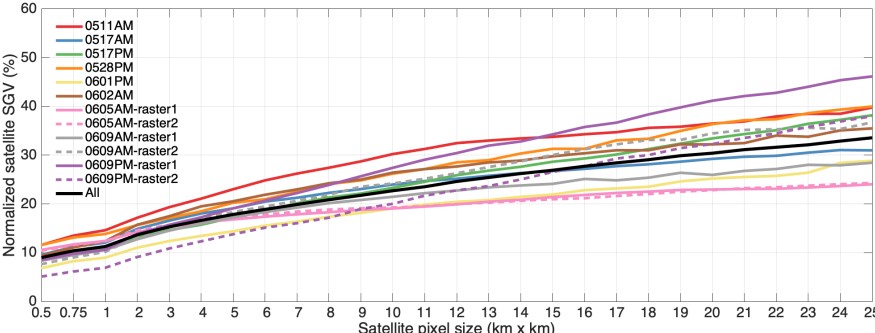

**Figure 5**. Average of the normalized satellite sub-grid variability (SGV) sampled individually
from the twelve rasters (represented by the colored lines), and sampled from all the twelve rasters
together (represented by the black line) over the Seoul Metropolitan Area during KORUS-AQ.
Normalized satellite SGV is calculated by the standard deviation of the GeoTASO data within the
sampled satellite pixel divided by the mean of the GeoTASO data within the sampled satellite
pixel.






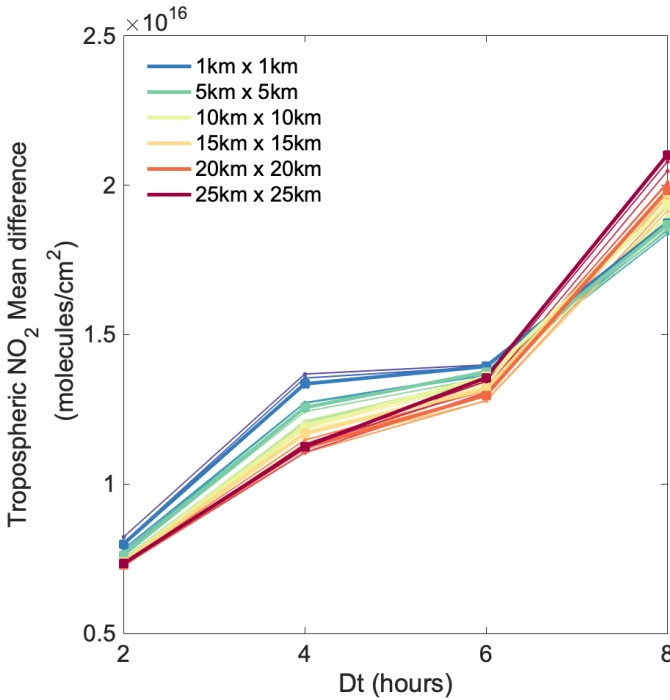

**Figure 6**. Temporal mean differences (TeMD) of hypothetical satellite pixel retrieved tropospheric
NO$_2$ vertical column (molecules cm$^{-2}$) over the Seoul Metropolitan Area (y-axis) as a function of
satellite pixel size time difference (Dt). Mean differences for the time difference of Dt are
calculated by averaging absolute value of the differences across all sampled satellite pixels that
have two values with time difference of Dt. Results for each pixel size are color-coded, with
selected sizes shown with thicker lines for reference.




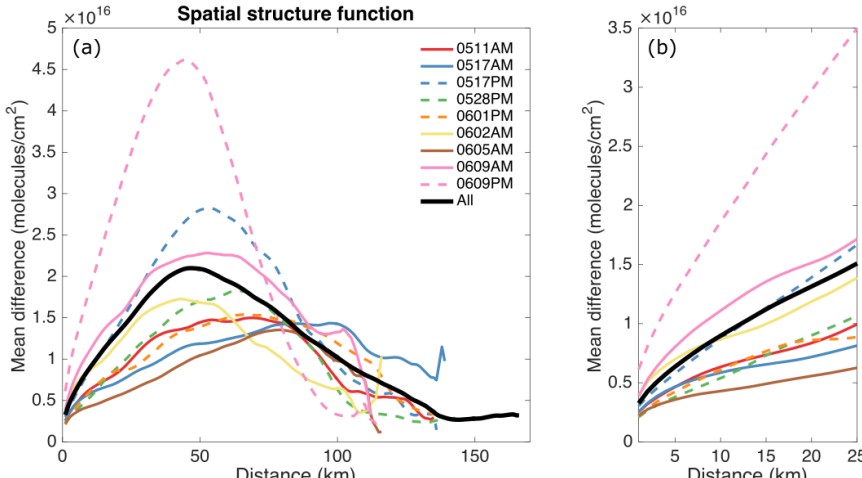

**Figure 7**. (a) Spatial Structure Function (SSF) for GeoTASO data of tropospheric $NO_2$ vertical
column molecules cm$^{-2}$) over the Seoul Metropolitan Area (SMA) during KORUS-AQ and (b) the
zoom-in version of panel (a) for distance range of 1-25 km. The SSF calculates average of absolute
value of $NO_{2,VC}$ differences (i.e., mean difference; y-axis) across all data pairs (measured in the
same hourly bin) that are separated by different distance (x-axis). The SSF based on GeoTASO
data measured during morning flights are in solid colored lines while the SSF based on GeoTASO
data measured during afternoon flights are in dashed colored lines. The SSF based on all the data
is in the black solid line.








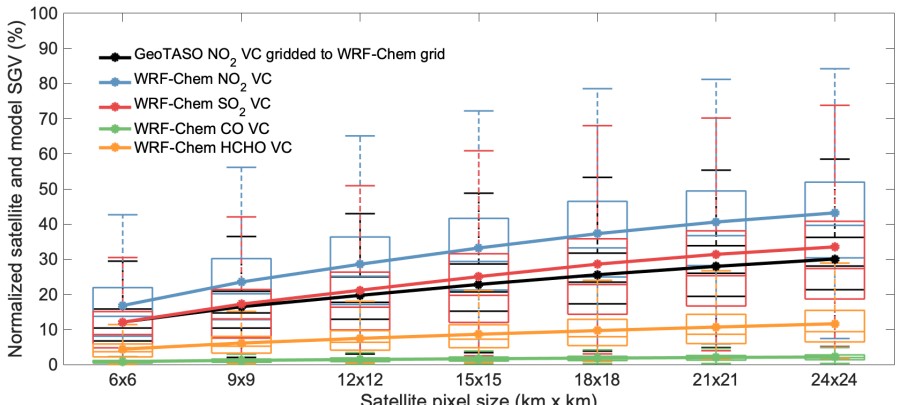

**Figure 8**. Boxplot of hypothetical satellite normalized SGV of $NO_2$ vertical column (VC), $SO_2$ VC, CO VC, and formaldehyde (HCHO) VC derived from the WRF-Chem simulation with a resolution of 3 km × 3 km (colored lines), and GeoTASO $NO_2$ VC that gridded to the WRF-Chem grid (black lines) over the Seoul Metropolitan Area. Medians are represented by red bars, interquartile ranges between 25th and 75th percentiles by blue boxes, and the most extreme data points not considered outliers by whiskers. The modeled $NO_2$, CO, $SO_2$, and HCHO are filtered to match the rasters of GeoTASO measurements.