# Peer review of "1Assessing sub-grid variability within satellite pixels over urban regions using2airborne mapping spectrometer measurements"

_Atmospheric Measurement Techniques, 2020_

## Referee Comment (RC1)

**Review of: Assessing sub-grid variability within satellite pixels using airborne mapping spectrometer measurements (Tang et al., 2021)**

The manuscript discusses sub-grid variability (SGV) within satellite pixels using high resolution airborne GeoTASO observations, acquired over three urban areas. A quantitative way to assess SGV for different hypothetical satellite pixel sizes is presented based on two methods: random sampling and spatial structure function. Also temporal variability in satellite pixels has been studied for different sizes. The scientific content of the paper fits well within the scope of AMT. However, major revisions (detailed below) need to be conducted in the paper before publication.

**General comments:**

-I repeat the comment from the quick access report that the whole document should be carefully scanned by the main author and some co-authors to remove unclarities and confusion. It is not always written in a well-thought concise way, hiding sometimes the key messages. Also, the authors refer often to "this error", "that method", etc. while it is not always clear to what is referred exactly. This effort would certainly bring the manuscript to a higher level.

-SGV is obviously strongly linked to effective variability/heterogeneity in the $NO_2$ field. It is shown that SGV is similar in three different studied regions. It is also stated that the method is generally applicable to quantify SGV and that for example a LUT could be used (e.g. p.3, l.94 and l.98). I doubt the latter. Even if the study is applied on three regions and even if you take 10.000 pixels randomly in this area, all of them are strongly urbanized which is reflected in the high mean $NO_2$ VC for all three areas. It should be clearly stated (in introduction and conclusion, maybe also by adapting the title) that results are valid for urban regions and are not proven for background areas that are characterized by much less heterogeneity. For such areas we can expect a lower impact of SGV. You could also specify that the SGV studied here can be seen as an upper limit, in the same way as you discussed for other species when compared to $NO_2$.

-The introduction doesn't mention anything about your study of temporal variability, while this is an important part in the further discussion and conclusion. Please write a few lines in the introduction and clarify why it is also relevant or an added-value to study the temporal variability in addition to SGV.

-p.2 l.66: "Until recently, accurate in-situ measurements with sufficient spatiotemporal coverage have not been available"→ I don't agree with this statement. For example, in the US and Europe there is a dense network of in-situ stations for quite a long time. The key message in this paragraph should be the difficulty to use in-situ observations for satellite validation and study of SGV or difficulty to compare a point measurement with gridded data in general and associated representativeness error. Please elaborate on this.

-p.2 l.78: Linked to the previous remark, start this paragraph by discussing the added-value of airborne mapping observations in general for SGV and satellite validation. There are other data sets used in other studies for assessing SGV (note the references made a bit earlier). Then, continue discussing the performed GEOTASO measurements used in this study.

-p.7 l. 268: Please reconsider if this paragraph is really needed here. It is rather confusing. Especially as temporal variability is discussed in the next section.

-Sect.4, paragraph 2: This paragraph is very confusing. It should be rewritten in a well thought and concise way. Comparison of satellite retrievals with aircraft CO vertical profiles are mentioned as motivation of

this work. Than it is mentioned that a same problem arises for $NO_2$ while you are mainly referring to satellite validation papers based on airborne data. Airborne mapping observations, able to cover full satellite pixels, are exactly a type of measurement that minimize the spatial representativeness error, this in contrast to in-situ, ground-based observations, vertical soundings, etc. Then, the next sentence starts to discuss other species and that GEOTASO is able to address 'this' problem (which problem?). GEOTASO was also used in the Nowlan et al. and Judd et al. references that were just mentioned and identified as: difficult to address 'this issue'.

-Sect. 3.2: Nothing is mentioned for conditions where Dt is larger than 4 hours. The case seems to be inverted from Dt 4 to 8 hours according to Figure 6 (TeMD increases with decreasing spatial resolution). Please add an explanation to the paper.

-Sect. 4: Please consider a discussion on the impact of your findings (both spatial and temporal variability) on top-down emission estimations from satellite observations at higher spatial resolution (e.g. TROPOMI, GEMS) and higher temporal resolution (e.g. GEMS, TEMPO, S-4), when compared to emission estimations from e.g. OMI?

**Minor comments:**

-p.3 l.85: 'As such, the GeoTASO data…' → Mention the GeoTASO spatial resolution already here.

-p.4 l.132: 'The dense sampling of the GeoTASO datasets is a unique feature' → There are quite a lot of other imaging systems that can obtain a similar or better spatial resolution. Please generalize this statement to overall airborne mapping observations.

-p.4 l.136: Which uncertainty estimate are you referring to?

-Sect. 2.1: It would help the reader to indicate a typical flight time to acquire one grid map already here in Sect. 2.1 in order to understand the temporal variability issue mentioned later on.

-p.6 l.222: In principle you still scan an area on the ground, so specify that you take the center lat/long of the pixels…or do I misunderstand this approach?

-p.8 l.297: You focused on GEO missions so far, so please mention as well S-4.

-p.8 l. 306: Not clear what you mean with "We also tested the results for sampling satellite pixels by raster instead of within hourly bins." …and what the difference is with the previous statements.

-Figure 6: Please rewrite caption in a more clear and concise way.

**Technical corrections:**

-p.6 l.225: Adapt "Distance" to "D"

-p.9 l.342: "…$NO_2$'s relatively short lifetime…" → …the relative short lifetime of $NO_2$…

-p.9 l.364: Correct the sentence.

-p.12 l. 458: HCHO has already been defined earlier.

-p.12 l. 468: Define 'local observations'. Do you mean ground-based in-situ?

-p.13 l. 503: This statement is not valid for TeMD > 4 hours.

---

## Referee Comment (RC2)

**Review of Tang et al. -- Assessing sub-grid variability within satellite pixels using airborne mapping  spectrometer measurements**

The authors investigate the variability of NO2 within hypothetical satellite footprint sizes based on high spatial resolution airborne imaging datasets. For this purpose, two different methods (random pixel sampling and spatial structure functions) are applied that provide consistent results. In addition, the authors also address temporal variability. The topic is of importance for the scientific community and fits well into the scope of AMT. However, I miss some information on the airborne dataset that should be included before publication.

**General comments**

- There is no explanation on the L2 retrieval of the NO2 tropospheric VCD.. I think that you need to briefly describe, or at least reference to the retrieval settings. The auxiliary data used in the L2 retrieval also impacts on the observed variability (e.g. NO2 vertical profile shape, surface reflectance)
  - What input data was used for the airborne measurements?
  - Did you use a consistent retrieval for all regions (SMA, Busan, LA)
- I think that the study on temporal variability would benefit from consideration of the wind conditions (speed, direction, variability)

**Detailed comments & technical corrections:**

| Page | Line | Comment |
|---|---|---|
| 0 | 0 | What is the meaning of the bold polygon in Fig 1, Fig 2, Fig S1? |
| 2 | 65 | also mention that comparison to in-situ observations is also difficult due to imperfect knowledge of the vertical profile |
| 2 | 75 | there are several more airborne instruments, which provide similar datasets. So these measurements are not really unique, see also P4.L126 |
| 4 | 130 | Remove square brackets from citation |
| 5 | 165 | You state that you sample 10k hypothetical pixels. However, considering the area covered by the flights and pixel sizes of up to 25km$^2$ it is not clear to me how many distinct samples are actually evaluated |
| 5 | 167 | Does -> do |
| 5 | 169 | How do you treat overlappings swaths from adjacent flight tracks? Do you also account for temporal differences between these overpasses? |
| 6 | 222 | The SSF ais defined here follows Follette-Cook et al. (2015) |
| 6 | 234 | "SMA in the Discussion section"… Please include a proper cross-reference |
| S | Fig S3 | The labels are way too small. Please increase the font size or reduce white space between subplots |
| 7 / S | 250/Fig S3 | The differences between median and mean values seem to be much larger for the SMA region that for Busan and LA. Please discuss possible reasons and the impact on the normalized SGV |

| | | |
|---|---|---|
| **7** | 257 | "discussed below". Please include a proper cross-reference. |
| **S6** | Fig S6 | Add a legend to the figure.
Mention in the caption that "red" corresponds to low values, and "blue" to high values. Consider using different colors, because "red" is also the color for the median |
| **7** | 265 | What is "this relationship" |
| **S** | S7 | Add a legend to the figure |
| **8** | 290ff | I am not sure if the threshold of ~10km spatial resolution can be generalized. It may be true for the regions investigated here. However, the spatial distribution of the NO2 field is also stongly affected by the meteorological conditions (strong winds lead to confined plumes, calm winds to high pollution levels above the sources) as well as the spatial distribution of the sources. |
| **9** | 341 | Why does a thicker PBL lead to stronger horizontal dispersion? |
| **9** | 348 | What about changing wind directions?  A change in wind direction would also lead to a shifted spatial pollution pattern, which consequently leads to a change in pollution levels over time above a certain location. |
| **10** | 373 | In Fig 6 you describe the increase of the mean differences of NO2 VCD with increasing time for the SMA region. The data over LA (Fig S11) does not show a similar behavior. Instead there is almost no change between Dt=4h and Dt=8h. Please provide possible explanations. |
| **10** | 349 | SSP? Do you mean SSF? |
| **10** | 406 | Are wind speeds of ~5m/s also representative for the measurement conditions of this study? |
| **11** | 425ff | What would be the dimensions of such a lookup table? Would you also consider the size of the city, the distribution of sources....? |
| **14** | 546ff | The reference styles are inconsistent.

* Some refereces have a DOI, others do not

* Some references have DOI as as clickable (blue) link, others do not

* Some refernces use doi:xxxxx, most others use https://doi.org/xxxx |

---

## Author Comment (AC1)

**Reviewer 1:**

The manuscript discusses sub-grid variability (SGV) within satellite pixels using high resolution airborne GeoTASO observations, acquired over three urban areas. A quantitative way to assess SGV for different hypothetical satellite pixel sizes is presented based on two methods: random sampling and spatial structure function. Also temporal variability in satellite pixels has been studied for different sizes. The scientific content of the paper fits well within the scope of AMT. However, major revisions (detailed below) need to be conducted in the paper before publication.

**Response:** Thank you for your time and effort in reviewing our manuscript. We have addressed the comments accordingly. Please see below for details.

**General comments:**

1. I repeat the comment from the quick access report that the whole document should be carefully scanned by the main author and some co-authors to remove unclarities and confusion. It is not always written in a well-thought concise way, hiding sometimes the key messages. Also, the authors refer often to "this error", "that method", etc. while it is not always clear to what is referred exactly. This effort would certainly bring the manuscript to a higher level.

**Response:** We have carefully revised the whole manuscript to convey our key messages more clearly. Please see the updated manuscript for details.

2. SGV is obviously strongly linked to effective variability/heterogeneity in the NO2 field. It is shown that SGV is similar in three different studied regions. It is also stated that the method is generally applicable to quantify SGV and that for example a LUT could be used (e.g. p.3, l.94 and l.98). I doubt the latter. Even if the study is applied on three regions and even if you take 10.000 pixels randomly in this area, all of them are strongly urbanized which is reflected in the high mean NO2 VC for all three areas. It should be clearly stated (in introduction and conclusion, maybe also by adapting the title) that results are valid for urban regions and are not proven for background areas that are characterized by much less heterogeneity. For such areas we can expect a lower impact of SGV. You could also specify that the SGV studied here can be seen as an upper limit, in the same way as you discussed for other species when compared to NO2.

**Response:** We thank the reviewer for pointing this out. We have changed the abstract, introduction, results, discussion, and conclusion sections to emphasize that the three regions analyzed in this study are urban regions, and the results are not tested over background areas that are characterized by much less heterogeneity. We also changed the manuscript title to include "over urban regions". Please see the updated manuscript for details.

3. The introduction doesn't mention anything about your study of temporal variability, while this is an important part in the further discussion and conclusion. Please write a few lines in the introduction and clarify why it is also relevant or an added-value to study the temporal variability

in addition to SGV.

**Response:** We added the following sentences to Paragraph 3 in the introduction:

"Temporal variability of satellite pixels is also an important issue in satellite design, validation, and application. For polar-orbiting satellites, knowledge of temporal variability is necessary to analyze the representativeness of satellite retrievals at specific overpass times. For geostationary Earth orbit (GEO) satellites, developing a measure of the temporal variability of fine-scale spatial structure will be important for assessing coincidence during validation of the new hourly observations."

We also added the following sentences to Paragraph 5 in the introduction:

"We use the tropospheric $NO_2$ vertical column (VC) retrieved by GeoTASO as a tool to assess satellite SGV and temporal variability for different hypothetical satellite pixel sizes over urban regions. Because spatial SGV and temporal variability both vary with satellite pixel size, the two need to be considered together to enhance the accuracy of satellite product analyses."

4. p.2 l.66: "Until recently, accurate in-situ measurements with sufficient spatiotemporal coverage have not been available" ➔ I don't agree with this statement. For example, in the US and Europe there is a dense network of in-situ stations for quite a long time. The key message in this paragraph should be the difficulty to use in-situ observations for satellite validation and study of SGV or difficulty to compare a point measurement with gridded data in general and associated representativeness error. Please elaborate on this.

**Response:** We agree with the reviewer that there are dense networks of in-situ atmospheric composition observations that can be used for satellite validation/evaluation (such as the EPA Air Quality System (AQS) and Aerosol Robotic Network (AERONET)). However, the requirements for satellite validation/evaluation and quantification of satellite SGV are different. To quantify satellite SGV, it requires dense observations within a satellite pixel. To our knowledge, such in-situ measurements are still very limited. To make the statement clearer, we changed the sentence to "Quantification of satellite SGV has historically been limited by insufficient spatial coverage of in situ measurements, and is a key issue...".

We also agree that it is sometimes challenging to use in-situ observations for satellite validation, and to compare a point measurement with gridded data in general as well as to address associated representativeness errors. Thus, we added the following statement to the paragraph:

"SGV introduces large uncertainties on top of the existing uncertainty introduced by imperfect knowledge of the trace gas vertical profiles. Accurate quantification of satellite SGV can facilitate the estimate of sampling uncertainty for satellite product validation/evaluation.".

5. p.2 l.78: Linked to the previous remark, start this paragraph by discussing the added-value of airborne mapping observations in general for SGV and satellite validation. There are other data sets used in other studies for assessing SGV (note the references made a bit earlier). Then, continue discussing the performed GEOTASO measurements used in this study.

**Response:** We have revised the paragraph by adding discussions of the added-value of airborne mapping spectrometer measurements in general for SGV, and the usage of data sets used in other

studies for assessing SGV. The revised paragraph is as follows:

"Airborne mapping spectrometer measurements provide dense observations within the several-kilometer footprint of a typical satellite pixel. This feature of airborne mapping spectrometer measurements provides a unique opportunity to estimate satellite SGV in addition to their role in satellite validation. For example, Broccardo et al. (2018) used aircraft measurements of $NO_2$ from an imaging differential optical absorption spectrometer (iDOAS) instrument to study intra-pixel variability in satellite tropospheric $NO_2$ column over South Africa, whilst Judd et al. (2019) evaluated the impact of spatial resolution on tropospheric $NO_2$ column comparisons with in situ observations using the $NO_2$ measurements of the Geostationary Trace gas and Aerosol Sensor Optimization (GeoTASO). GeoTASO is an airborne remote sensing instrument capable of high spatial resolution retrieval of UV-VIS absorbing species like $NO_2$, formaldehyde (HCHO; Nowlan et al., 2018), and sulfur dioxide ($SO_2$; Chong et al., 2020), and with measurement characteristics similar to the GEMS and TEMPO GEO satellite instruments…"

6. p.7 l. 268: Please reconsider if this paragraph is really needed here. It is rather confusing. Especially as temporal variability is discussed in the next section.

**Response:** We deleted the paragraph.

7. Sect.4, paragraph 2: This paragraph is very confusing. It should be rewritten in a well thought and concise way. Comparison of satellite retrievals with aircraft CO vertical profiles are mentioned as motivation of this work. Than it is mentioned that a same problem arises for NO2 while you are mainly referring to satellite validation papers based on airborne data. Airborne mapping observations, able to cover full satellite pixels, are exactly a type of measurement that minimize the spatial representativeness error, this in contrast to in-situ, ground-based observations, vertical soundings, etc. Then, the next sentence starts to discuss other species and that GEOTASO is able to address 'this' problem (which problem?). GEOTASO was also used in the Nowlan et al. and Judd et al. references that were just mentioned and identified as: difficult to address 'this issue'.

**Response:** We have revised the paragraph to remove potential confusion. We also would like to clarify that although GeoTASO data minimizes the spatial representativeness error (which is in contrast to in-situ measurements), we do not emphasize the advantages of GeoTASO over other in situ measurements for satellite validation/evaluation purposes, or suggest replacing other in situ measurements with GeoTASO for satellite validation/evaluation. Instead, the goal of this study is using GeoTASO to provide SGV estimates that can serve as a useful reference for the comparison between satellite retrievals and in situ measurements that have representativeness errors. This statement is included in the conclusion (Section 5). We changed "aircraft profiles" to "in situ measurements" in this paragraph. We changed "this problem" to "the issue of satellite SGV and representativeness error of in situ measurements in satellite validation/evaluation". The updated paragraph is:

"Previous studies recognized the challenges in satellite validation/evaluation for $NO_2$ retrievals due to satellite SGV and representativeness error of in situ measurements (e.g., Nowlan et al., 2016, 2018; Judd et al., 2019; Pinardi et al., 2020; Tack et al., 2020). The gapless airborne mapping datasets of GeoTASO with sufficient spatiotemporal resolution are a promising way to address the

issue of satellite SGV and representativeness errors in satellite validation/evaluation (e.g., Nowlan et al., 2016, 2018; Judd et al., 2019).

"Challenges due to SGV also have implications for other trace gas column measurements. For example, in Tang et al. (2020), satellite SGV and representativeness errors of in situ measurements introduced uncertainties in validation of CO retrievals from the MOPITT (Measurement Of Pollution In The Troposphere) satellite instrument. Normalized SGV of the GeoTASO tropospheric $NO_2$ VC might serve as an upper bound to the SGV of CO, $SO_2$ and other species that share common source(s) with $NO_2$ but with relatively longer lifetimes than $NO_2$, even if their spatial distributions have different patterns (e.g., Chong et al., 2020). For example, at the resolution of 22 km × 22 km (resolution of MOPITT CO retrievals), the expected normalized satellite SGV of tropospheric $NO_2$ VC is ~30%. Therefore, we might expect the normalized satellite SGV for tropospheric CO VC to be lower than this value.".

8. Sect. 3.2: Nothing is mentioned for conditions where Dt is larger than 4 hours. The case seems to be inverted from Dt 4 to 8 hours according to Figure 6 (TeMD increases with decreasing spatial resolution). Please add an explanation to the paper.

**Response:** We added the following explanation:

"As the time difference Dt increases, the temporal variability TeMD increases for all pixel sizes. However, the TeMD is now greater at large pixel size which is in contrast to the higher TeMD at small pixel size for shorter Dt. This is a result of the pollution pattern that develops over the SMA during the day (June $9^{th}$, 2019) as described above. The higher TeMD reflects the fact that many of the large pixels now span the strong $NO_2$ gradient between the urban and surrounding area resulting in a much higher spatial variability than earlier in the day at a spatial scale not captured with the smaller pixels. As a caution, we note that TeMD for 8 hours is determined by only the difference between Raster 1 of the 0609AM and Raster 2 of 0609PM (Figure 2), and that the regional coverage for Raster 2 of 0609PM is different from the coverage of the other PM rasters. Therefore, the relationship of TeMD and spatial resolution for a large Dt (e.g., 6 or 8 hours) over SMA requires further study.".

9. Sect. 4: Please consider a discussion on the impact of your findings (both spatial and temporal variability) on top-down emission estimations from satellite observations at higher spatial resolution (e.g. TROPOMI, GEMS) and higher temporal resolution (e.g. GEMS, TEMPO, S-4), when compared to emission estimations from e.g. OMI?

**Response:** We added the following paragraph to Section 4:

"For data assimilation and inverse modeling application (e.g., top-down emission estimations from satellite observations), it is essential to accurately characterize the observation error covariance matrix **R** (Janjíc et al., 2017). The first component of **R** is the instrument error covariance matrix due to instrument noise and retrieval uncertainty in the case of trace gas satellite data. The second component is the representation error covariance matrix, arising from fundamental differences of the atmospheric sampling, typically when assimilating a local point measurement into a grid-based model (Boersma et al, 2016). The observation error covariance due to representativeness error is difficult to define, but can be parameterized when calculating super observations by inflating the

observation error variances (Boersma et al., 2016) and quantified by a posteriori diagnostics estimation (Gaubert et al. 2014). Knowledge of the fine-scale model sub-grid variability is therefore essential to verify those assumptions and inform error statistics for application to chemical data assimilation studies. Our results suggest large potential improvements in emission estimates when assimilating high spatial resolution TROPOMI and GEO satellite data with SGV of ~10%–20% (Figure 4), compared to OMI data with SVG of ~30% (Figure 4), in line with the existing literature for $NO_2$ (e.g., Valin et al., 2011). We have also shown that significant temporal variability of $NO_2$ is expected at higher spatial resolutions. This observed signal will open new avenue for space-based monitoring of atmospheric chemistry and will reduce errors of inverse estimates of fluxes."

- Boersma, K. F., Vinken, G. C. M., and Eskes, H. J.: Representativeness errors in comparing chemistry transport and chemistry climate models with satellite UV–Vis tropospheric column retrievals, Geosci. Model Dev., 9, 875–898, https://doi.org/10.5194/gmd-9-875-2016, 2016.
- Gaubert, B., Coman, A., Foret, G., Meleux, F., Ung, A., Rouil, L., Ionescu, A., Candau, Y., and Beekmann, M.: Regional scale ozone data assimilation using an ensemble Kalman filter and the CHIMERE chemical transport model, Geosci. Model Dev., 7, 283–302, https://doi.org/10.5194/gmd-7-283-2014, 2014.
- Janjic, T., Bormann, N., Bocquet, M., Carton, J. A., Cohn, ´S. E., Dance, S. L., Losa, S. N., Nichols, N. K., Potthast, R., Waller, J. A., and Weston, P.: On the representation error in data assimilation, Q. J. R. Meteorol. Soc., 144, 1257–1278, https://doi.org/10.1002/qj.3130, 2018.
- Valin, L. C., Russell, A. R., Hudman, R. C., and Cohen, R. C.: Effects of model resolution on the interpretation of satellite NO2 observations, Atmos. Chem. Phys., 11, 11647–11655, https://doi.org/10.5194/acp-11-11647-2011, 2011.

**Minor comments:**

10. p.3 l.85: 'As such, the GeoTASO data…' ➔ Mention the GeoTASO spatial resolution already here.

**Response:** We added "(with a spatial resolution of ~250 m × 250 m)" to the sentence.

11. p.4 l.132: 'The dense sampling of the GeoTASO datasets is a unique feature' ➔There are quite a lot of other imaging systems that can obtain a similar or better spatial resolution. Please generalize this statement to overall airborne mapping observations.

**Response:** We changed it to "the dense sampling of airborne remote sensing measurements such as GeoTASO is a unique feature".

12. p.4 l.136: Which uncertainty estimate are you referring to?

**Response:** It refers to the validation results of GeoTASO VC $NO_2$ retrievals during KORUS-AQ. We expanded the description of uncertainty estimate of GeoTASO $NO_2$ retrievals in Section 2.1. Specifically, we changed

"Validation of GeoTASO $NO_2$ retrievals during KORUS-AQ with Pandora shows ~10% difference on average. The uncertainty estimate is lower than that reported by Nowlan et al. (2016)."

to

"GeoTASO $NO_2$ VC retrievals have been validated with aircraft in situ data and ground-based Pandora remote sensing measurements during KORUS-AQ. Validation of GeoTASO $NO_2$ VC retrievals with aircraft in situ data suggest ~25% average difference, while agreement with Pandora is better with a difference of ~10% on average. Mean difference between Pandora and aircraft in situ data is ~20%. These validation results of GeoTASO $NO_2$ VC retrievals are better than that reported by Nowlan et al. (2016). GeoTASO $NO_2$ VC retrievals during 2017 SARP have also been validated with Pandora data (Judd et al., 2019)."

13. Sect. 2.1: It would help the reader to indicate a typical flight time to acquire one grid map already here in Sect. 2.1 in order to understand the temporal variability issue mentioned later on.

**Response:** We add the statement "It took ~4 hours to sample the large-area rasters (i.e., 0511AM, 0517AM, 0517PM, 0528PM), and ~2 hours to sample small-area rasters (i.e., 0601PM, 0602AM, 0605AM, 0609AM, and 0609PM)" in Section 2.1.

14. p.6 l.222: In principle you still scan an area on the ground, so specify that you take the center lat/long of the pixels...or do I misunderstand this approach?

**Response:** The locations of the GeoTASO pixel centers are used to calculate the distances. We added this information in the manuscript.

15. p.8 l.297: You focused on GEO missions so far, so please mention as well S-4.

**Response:** Thank you for pointing this out. We added "Sentinel-4" to the list of the satellite instruments in the sentence.

16. p.8 l. 306: Not clear what you mean with "We also tested the results for sampling satellite pixels by raster instead of within hourly bins." ...and what the difference is with the previous statements.

**Response:** The GeoTASO data located closely in space may be sampled at slightly different times for the same flight. The paragraph aims to provide a discussion on the possible impacts of grouping/aggregating data samples at different time frequency (e.g., every entire flight or every entire raster or every hour) in our analyses of spatial SGV. To make it clearer, we revised the paragraph:

"We tested the sensitivity of the results over SMA to sampling GeoTASO data with hypothetical satellite pixels grouped by complete flight, rather than grouping the data by time in hourly bins.

The resulting patterns and relationships are similar, except that the normalized satellite SGV increases ~5% for pixels of small sizes due to the inclusion of temporal variability (Figure S8a). We also tested the results for sampling satellite pixels by raster instead of within hourly bins. The results are again similar to Figure 4, except that the normalized satellite SGV increases ~1% for pixels of small sizes due to the inclusion of temporal variability (Figure S8b)."

to

"The GeoTASO data located closely in space may be sampled at slightly different times for the same flight. To explore the impact of temporal variability on this SGV analysis, we performed two sensitivity tests. The typical time period for a complete flight is ~4 hours. In the first test, we sampled GeoTASO data with hypothetical satellite pixels grouped by each complete flight, rather than grouping the data by each hour (i.e., hourly bins). The resulting patterns and relationships are similar to those derived from grouping data into hourly bins, except that the normalized satellite SGV increases ~5% for small pixels due to temporal variability (Figure S7a). In the second test, we sampled GeoTASO data with hypothetical satellite pixels grouped by each raster. The results are still similar to those derived from grouping data into hourly bins (Figure 4), except that the normalized satellite SGV increases ~1% for small pixels due to the inclusion of temporal variability (Figure S7b). This is because sampling by raster includes smaller temporal variability than sampling by flight, but larger temporal variability than sampling by hourly bins.".

17. Figure 6: Please rewrite caption in a more clear and concise way.

**Response:** We revised the caption of Figure 6.

**Technical corrections:**

18. p.6 l.225: Adapt "Distance" to "D"

**Response:** We changed "Distance" to "D".

19. p.9 l.342: "…NO2's relatively short lifetime…" ➔ …the relative short lifetime of NO2…

**Response:** We made the change.

20. p.9 l.364: Correct the sentence.

**Response:** We changed the sentence to "This is expected because averaging over a larger region smooths out temporal variability, and therefore produces smaller hourly differences".

21. p.12 l. 458: HCHO has already been defined earlier.

**Response:** We deleted the full name of HCHO in the sentence.

22. p.12 l. 468: Define 'local observations'. Do you mean ground-based in-situ?

**Response:** We changed "local observations" to "in situ observations" in the manuscript.

23. p.13 l. 503: This statement is not valid for TeMD > 4 hours.

**Response:** Thank you for pointing this out. We changed the statement to "Temporal variability (TeMD) increases when increasing the satellite retrieval spatial resolution (i.e., smaller pixel size) in SMA when time difference is small (Dt<=4 hours)".

---

## Author Comment (AC2)

**Reviewer 2:**

The authors investigate the variability of NO2 within hypothetical satellite footprint sizes based on high spatial resolution airborne imaging datasets. For this purpose, two different methods (random pixel sampling and spatial structure functions) are applied that provide consistent results. In addition, the authors also address temporal variability. The topic is of importance for the scientific community and fits well into the scope of AMT. However, I miss some information on the airborne dataset that should be included before publication.

**Response:** Thank you for your time and effort in reviewing our manuscript. We have addressed the comments accordingly. Please see below for details.

**General comments**

1. There is no explanation on the L2 retrieval of the NO2 tropospheric VCD. I think that you need to briefly describe, or at least reference to the retrieval settings. The auxiliary data used in the L2 retrieval also impacts on the observed variability (e.g. NO2 vertical profile shape, surface reflectance)

  o What input data was used for the airborne measurements?

  o Did you use a consistent retrieval for all regions (SMA, Busan, LA)

**Response:** We added the following description to Section 2.1 of the manuscript:

"$NO_2$ is retrieved from GeoTASO spectra using the Differential Optical Absorption Spectroscopy (DOAS) technique. The retrieval methods and Level 2 data processing are described in Lamsal et al. (2017) and Souri et al. (2020) for KORUS-AQ and in Judd et al. (2019) for SARP. Although beyond the scope of this work, it is important to recognize that assumptions made in the retrieval process (e.g., assumed vertical distribution of the $NO_2$ profile) could affect the final variability of the retrieved $NO_2$ fields."

2. I think that the study on temporal variability would benefit from consideration of the wind conditions (speed, direction, variability)

**Response:** Thank you for pointing this out. We analyzed wind fields from MERRA-2 reanalysis dataset for the 12 rasters over Seoul Metropolitan Area and 5 rasters in the LA Basin. We found that the wind conditions could be used to partially explain the patterns of the temporal variability, even though the impact of wind conditions on spatial variability and spatial SGV is small. We have included wind patterns of the MERRA-2 model surface level and 3 km (a.s.l.) in the supplement, and added discussion in the manuscript. Specifically, we have included the following statement to the manuscript:

"In addition, changing wind conditions (speed and direction; Figure S9) during the day can also lead to a shift in pollution pattern, and result in different pollution conditions for the same pixel at different time of a day. For example, Raster 1 of the 0609AM (9.17 local time) and Raster 2 of 0609PM (17 local time) are used to calculate TeMD for Dt equals 8 hours. The differences in wind conditions (Figure S9) and the pollution patterns (Figure 2) are large.".

**Detailed comments & technical corrections:**

3. Page 0 Line 0: What is the meaning of the bold polygon in Fig 1, Fig 2, Fig S1?

**Response:** The bold polygon represents the political boundary of Seoul. We added this information in the captions of Fig 1, Fig 2, and Fig S1.

4. Page 2 Line 65: also mention that comparison to in-situ observations is also difficult due to imperfect knowledge of the vertical profile

**Response:** We added the statement in the manuscript:

"… SGV introduces large uncertainties on top of the existing difficulty due to imperfect knowledge of the vertical profiles."

5. Page 2 Line 75: there are several more airborne instruments, which provide similar datasets. So these measurements are not really unique, see also P4.L126.

**Response:** We removed the adjective "unique" for GeoTASO in the manuscript.

6. Page 4 Line 130: Remove square brackets from citation

**Response:** We changed all the square brackets to parentheses for citation.

7. Page 5 Line 165: You state that you sample 10k hypothetical pixels. However, considering the area covered by the flights and pixel sizes of up to 25km2 it is not clear to me how many distinct samples are actually evaluated.

**Response:** Because we discarded a sampled satellite pixel if it is not covered by GeoTASO data for at least 75% of its area, the actual distinct sample sizes (~10% of all the samples) are much smaller than 10,000. For each hourly bin, there are ~1000 samples, therefore for a flight (typically 4 hours), there are 3200 samples for one satellite pixel size. The samples are sufficient as our sensitivity test indicates that the results do not change by halving the sample size. This information is included in the manuscript (Section 2.4).

8. Page 5 Line 167: Does -> do

**Response:** This was corrected in the previous revision process (i.e., minor revision before posting as preprint for interactive discussion).

9. Page 5 Line 169: How do you treat overlappings swaths from adjacent flight tracks? Do you

also account for temporal differences between these overpasses?

**Response:** As the GeoTASO data located closely in space may be sampled at slightly different times for the same flight, we separate GeoTASO data into hourly bins for each flight before pixel sampling in order to reduce the impact of temporal variability of the GeoTASO data within a single satellite pixel sample.

In addition, to quantify possible impacts of temporal differences in aggregating/grouping data samples used for our analyses of spatial SGV, we did two sensitivity tests that include different levels of temporal differences in the satellite pixel random sampling for spatial variability process.

(1) We tested the sensitivity of the results over SMA to sampling GeoTASO data with hypothetical satellite pixels grouped by complete flight, rather than grouping the data by time in hourly bins. The resulting patterns and relationships are similar, except that the normalized satellite SGV increases ~5% for pixels of small sizes due to the inclusion of temporal variability (Figure S7a).

(2) We then also tested the results for sampling satellite pixels by raster instead of within hourly bins. The results are again similar to Figure 4, except that the normalized satellite SGV increases ~1% for pixels of small sizes due to the inclusion of temporal variability (Figure S7b).

10. Page 6 Line 222: The SSF ais defined here follows Follette-Cook et al. (2015)

**Response:** We changed "is" to "as".

11. Page 6 Line 234: "SMA in the Discussion section"… Please include a proper cross-reference

**Response:** We changed "the Discussion section" to "Section 4".

12. S Fig S3: The labels are way too small. Please increase the font size or reduce white space between subplots.

**Response:** We reduced white space between subplots and increased the font size for Figures S3-S7.

13. Page 7/S Line 250/Fig S3: The differences between median and mean values seem to be much larger for the SMA region that for Busan and LA. Please discuss possible reasons and the impact on the normalized SGV.

**Response:** The mean values are larger than median values over SMA, while over the other two regions, mean and median values are relatively close. This is likely due to the high pollution level and extreme pollution events over SMA. Overall, we do not expect this to have a significant impact on the normalized satellite SGV. Because the high pollution level and extreme pollution events over SMA also lead to higher standard deviation (SD) besides higher mean. Higher SD and higher mean cancel out in the calculation of normalized SGV (the standard deviation of the GeoTASO data within the sampled satellite pixel divided by the mean of the GeoTASO data within the sampled satellite pixel; $SD_{pixel}/MEAN_{pixel}$). This is also consistent with our results– the pattern of

normalized SGV over the three regions are similar, even though they have different levels of pollution (Figure 4).

14. Page 7 Line 257: "discussed below". Please include a proper cross-reference.

**Response:** We deleted the statement "that is discussed below".

15. S6 Fig S6 Add a legend to the figure.

Mention in the caption that "red" corresponds to low values, and "blue" to high values. Consider using different colors, because "red" is also the color for the median.

**Response:** We added the legend.

When there are multiple boxplots in one panel (i.e., Figures S6 and S7), the color for the median is the same as the color of the box instead of red. For example in Figure S6, median of morning data is blue while the median of afternoon data is red). Therefore, there is no need to change to a different color.

16. Page 7 Line 265: What is "this relationship"

**Response:** We deleted the paragraph/sentence.

17. S S7 Add a legend to the figure

**Response:** We deleted the figure.

18. Page 8 Line 290ff: I am not sure if the threshold of ~10km spatial resolution can be generalized. It may be true for the regions investigated here. However, the spatial distribution of the NO2 field is also stongly affected by the meteorological conditions (strong winds lead to confined plumes, calm winds to high pollution levels above the sources) as well as the spatial distribution of the sources.

**Response:**

We analyzed wind fields from MERRA-2 reanalysis dataset for the 12 rasters (new Figure S9), and found that the wind conditions (both wind speed and wind direction) vary strongly among the 12 rasters. Related to the wind conditions, the spatial distributions of $NO_2$ field and pollution levels above the sources also vary strongly among the 12 rasters. We agree with the reviewer that the wind conditions can affect the spatial distribution of the $NO_2$ field and pollution levels above the sources. However, as we note in the manuscript (Section 3.1), the results show that the normalized SGV is not affected by pollution levels, and therefore less likely to be affected by wind field.

Nevertheless, we deleted the sentence to avoid potential confusion.

19. Page 9 Line 341: Why does a thicker PBL lead to stronger horizontal dispersion?

**Response:** During the daytime, increasing surface temperature leads to stronger vertical mixing and hence greater PBL height. In general, the greater vertical mixing is associated with stronger horizontal divergence at the top of the convective cell within PBL and hence potentially a stronger horizontal dispersion due to the divergence. We have revised the statement in the manuscript to reflect the explanation:

"As the day progresses, the PBL height increases (~1800 m during 15:00-17:00; Figure S9) allowing for greater horizontal mixing to take place."

to

"As the day progresses, the PBL height increases (~1800 m during 15:00-17:00; Figure S9) due to enhanced convection, which further induces a stronger horizontal divergence at the top of the convective cell and hence allows for greater horizontal dispersion to take place along with the divergence.".

20. Page 9 Line 348: What about changing wind directions? A change in wind direction would also lead to a shifted spatial pollution pattern, which consequently leads to a change in pollution levels over time above a certain location.

**Response:** Please see the response to Reviewer 2, Comment 2.

21. Page 10 Line 373: In Fig 6 you describe the increase of the mean differences of NO2 VCD with increasing time for the SMA region. The data over LA (Fig S11) does not show a similar behavior. Instead there is almost no change between Dt=4h and Dt=8h. Please provide possible explanations.

**Response:** The data over the LA is limited. Besides the limited data, different wind conditions over SMA and LA could be a possible reason for the difference in TeMD. We added the following discussion in the manuscript:

"For the LA Basin GeoTASO data, sampled hypothetical satellite pixels show TeMD increases at higher spatial resolution for the available Dt equal to 4 and 8 hours (Figure S11). However, TeMD is fairly constant at these two time differences which is different to what was observed over SMA (Figure 6). We note that with only 2 flight days of flight data, the GeoTASO data over LA is also limited, which may be the main driver of the difference. Besides the limited data, one possible reason is the different wind fields over the two regions. As mentioned previously, Raster 1 of the 0609AM and Raster 2 of 0609PM are used to calculate TeMD for Dt equals 8 hours over SMA. The differences in wind direction (Figure S9) for the two rasters are large (almost opposite in some cases). However, over LA, the differences in wind direction (Figure S12) for the two rasters (Rasters 1 and 3 for 0627 flight) are relatively small, compared to the differences over SMA.".

22. Page 10 Line 349: SSP? Do you mean SSF?

**Response:** The typo was corrected in the previous revision process (i.e., minor revision before posting as preprint for interactive discussion).

23. Page 10 Line 406: Are wind speeds of ~5m/s also representative for the measurement conditions of this study?

**Response:** We analyzed wind fields from MERRA-2 reanalysis dataset for the 12 rasters (new Figure S9). The averaged wind speed over the SMA domain region (upper left box in Figure 1) for the 12 rasters vary from ~1 m/s to ~3 m/s at the model surface level, from ~1 m/s to ~10 m/s at 3 km, and from ~2 m/s to ~17 m/s at 5 km. ~5m/s is within the range of the conditions represented by the 12 rasters.

24. Page 11 Line 425ff: What would be the dimensions of such a lookup table? Would you also consider the size of the city, the distribution of sources….?

**Response:** The three cities we studied have different levels of pollution and urbanization, city sizes, and PBL conditions. Their normalized SGV have a similar a pattern, indicating the pattern of normalized SGV may be generalizable to $NO_2$ VC over regions with different levels of urbanization and air pollution, and different PBL conditions. Therefore, in our future study and development of the lookup table by including more campaign data, we do not expect to include these dimensions. However, if the future study suggests such lookup table is not generalizable, we will alternatively provide the statistics for normalized SGV as a function of potential driving factors such as levels of pollution, city sizes, meteorological and seasonal conditions.

25. Page 14 Line 546ff: The reference styles are inconsistent.

* Some refereces have a DOI, others do not

* Some references have DOI as as clickable (blue) link, others do not

* Some refernces use doi:xxxxx, most others use https://doi.org/xxxx

**Response:** We unified the reference format according to the EGU's guide for reference format.